# Hyperosmotic Stress Response Memory is Modulated by Gene Positioning in Yeast

**DOI:** 10.3390/cells8060582

**Published:** 2019-06-13

**Authors:** Zacchari Ben Meriem, Yasmine Khalil, Pascal Hersen, Emmanuelle Fabre

**Affiliations:** 1Université de Paris, Laboratoire Matière et Systèmes Complexes, CNRS UMR 7057, F-75013 Paris, France; zack_ben@hotmail.fr; 2Université de Paris, Laboratoire Génomes, Biologie Cellulaire et Thérapeutiques, CNRS UMR7212, INSERM U944, Centre de Recherche St Louis, F- 75010 Paris, France; khalil.yasmine.93@gmail.com

**Keywords:** chromosome organization, cellular memory, single cell, stress response, yeast

## Abstract

Cellular memory is a critical ability that allows microorganisms to adapt to potentially detrimental environmental fluctuations. In the unicellular eukaryote *Saccharomyces cerevisiae*, cellular memory can take the form of faster or slower responses within the cell population to repeated stresses. Using microfluidics and fluorescence time-lapse microscopy, we studied how yeast responds to short, pulsed hyperosmotic stresses at the single-cell level by analyzing the dynamic behavior of the stress-responsive *STL1* promoter (pSTL1) fused to a fluorescent reporter. We established that pSTL1 exhibits variable successive activation patterns following two repeated short stresses. Despite this variability, most cells exhibited a memory of the first stress as decreased pSTL1 activity in response to the second stress. Notably, we showed that genomic location is important for the memory effect, since displacement of the promoter to a pericentromeric chromatin domain decreased the transcriptional strength of pSTL1 and led to a loss of memory. This study provides a quantitative description of a cellular memory that includes single-cell variability and highlights the contribution of chromatin structure to stress memory.

## 1. Introduction

Cellular memory can be defined as a cellular response to transient and repeated stimuli. Constantly fluctuating and potentially stressful environments can induce cellular memory, and thus possibly exert selective pressure on cell viability [1]. Living organisms have developed various strategies to cope with environmental changes in order to ensure their survival. Regulation of gene transcription is one possible mechanism by which cells can maintain their biological functions within a challenging environment [2]. Active genetic responses that allow cells to survive a single stimulus are termed cellular adaptation. Factors such as post-translational modification of histones, chromatin remodeling, production of specific proteins during stress or even changes in chromatin conformation have been determined as causal factors involved in adaptation to environmental changes [3,4]. What happens when cells encounter consecutive stresses is less-well understood. However, in some cases, adaptation to an initial stress has been shown to serve as a learning process that results in better adaptation to subsequent stresses; this process is designated as cellular memory [5]. The biological mechanisms that underlie cellular memory include chromatin remodeling (epigenetic memory) or synthesis of proteins during the first stress; such proteins may leave a trace of the first event, which is termed cytoplasmic memory [6,7,8].

Studies of cellular memory in budding yeast have shown that cells can respond differently when confronted by successive environmental stresses. For instance, the so-called galactose memory is characterized by more rapid transcriptional reactivation of the GAL cluster, while the memory of long hyperosmotic stresses is characterized by reduced activation of the osmo-responsive gene *GRE2* without any temporal change in the reactivation of this gene [6,8]. 

All eukaryotes have a highly organized nucleus. Yeast chromosomes follow a Rabl organization; centromeres are tethered to the spindle pole body while telomeres are anchored to the nuclear periphery [9,10]. Interestingly, the galactose or inositol memories appear to rely on 3D gene positioning, since repositioning of the *INO1* gene or GAL cluster towards the nuclear periphery in an H2A.Z and nucleoporin-dependent manner is important for memory [8,11]. Nuclear organization may also play a critical role in the stress response as most stress response genes are located in subtelomeres. Subtelomeres lack essential genes, but are enriched in rapidly evolving non-essential gene families that are required to adapt to environmental change [12]. Subtelomeres are subjected to silencing by proteins of the silent information regulator (SIR) complex; however, stress conditions can inhibit this repression [13,14,15]. 

Most of the studies investigating memory effects have been performed on isogenic populations of cells, which only provide information on the mean behavior of the population [16]. However, cellular populations are heterogeneous due to extrinsic noise, such as the age, size or position of each cell in the cell cycle (for reviews, [17,18]). Moreover, gene expression is an inherently stochastic phenomenon due to the low number and limited availability of transcription factors and accessibility of the promoters or functional regulatory networks [19]. Overall, stochasticity causes genetically identical cells to exhibit variable behaviors when exposed to identical stimuli. 

The response of budding yeast to osmotic changes has proven a useful tool to study the emergence of adaptation and cellular memories in this organism [20,21]. When yeast face an increase in the osmolarity of the environment (hyperosmotic stress), intracellular water flows out of the cell, leading to cell shrinkage [22]. This imbalance in osmotic pressure is detected by osmosensors that activate the high osmolarity glycerol (HOG) pathway, which phosphorylates the cytoplasmic protein Hog1 [23]. Phosphorylated Hog1 translocates into the nucleus where it participates in the activation and regulation of an estimated 10% of the genome, including the osmo-responsive gene *STL1* [24]. The HOG pathway allows yeast to physiologically adapt to hyperosmotic stress within 15–30 min [25], mainly by producing glycerol to achieve homeostasis. Dephosphorylation and translocation of Hog1 out of the nucleus signal the end of the adaptation to hyperosmotic stress. 

Here, we present a single-cell study of *S*. *cerevisiae* exposed to short pulses of hyperosmotic stress in a well-controlled system based on time-lapse fluorescence microscopy and microfluidics [26,27]. Hundreds of single cells receiving repeated osmotic stresses were tracked and analyzed. In response to two consecutive hyperosmotic stresses separated by 4 h, individual cells displayed variability in the dynamic activity of pSTL1 in response to the first and second stress. Despite the existence of this pronounced dynamic variability, most cells exhibited the same behavior, namely, the response to the second stress was reduced in amplitude. We termed this specific behavior the memory effect. Importantly, we found that the chromatin environment modulates the cellular response to pulsed stresses. Relocation of the promoter of interest close to the centromere reduced the activity of pSTL1 and led to a loss of the memory effect. Overall, this study suggests that the specific location of pSTL1 at the subtelomere is necessary for the optimal level of transcription required to go beyond simple stochastic behavior and to enable the emergence of memory in response to short osmotic stresses.

## 2. Materials and Methods

### 2.1. Flow Cytometry

All flow cytometry experiments were performed using a Gallios flux cytometer (Beckman Coulter, Brea, CA, USA) equipped with ten colors and four lasers (488 nm blue, 561 nm yellow, 638 nm red, 405 nm violet). We used the 488 nm excitation laser and 530 ± 30 nm emission filter. 

### 2.2. Yeast Strains and Cell Culture 

Experiments were performed using a pSTL1::yECITRINE-His5 (yPH53 or YEF1093) strain derived from S288C (kindly gifted by Megan McClean, College of Engineering, University of Wisconsin-Madison, WI, USA). The yeast cells were grown overnight in synthetic complete medium (SC; 6.7 g/L yeast nitrogen base without amino acids, 2 g/L complete amino acids mix) containing 2% glucose at 30 °C. The next morning, the cells were diluted to OD_600_ = 0.5. 

The genotypes of all strains used in this study are indicated in Table A1. To move the pSTL1 reporter construct to the peri-centromeric region of chromosome IV, the pSTL1-yECITRINE-HIS5 construct was PCR amplified using the oZB6 and oZB7 primers (Table A2), which share a 50 base pair homology with the *TRP1* locus. [HIS+ TRP−] yeast transformants were verified by PCR and the PCR fragment was sequenced to confirm the absence of mutations in the construct. 

### 2.3. CRISPR-dCAS9 Experiments 

The plasmid pAG414GPD-dCas9-VPR from Addgene plasmid (# 63801) was used to express the inactivated form of CAS9 fused to the transcriptional activator VPR. The guides were cloned into the plasmid pEF534 under the control of the SNR52 promoter using the enzyme BsmBI. Marker exchange was conducted by digesting the resulting plasmid with NotI/XbaI and cloning the guide-containing fragment into pRS425 digested with NotI/XbaI. The two guides designed to target pSTL1 are gRNA1 and gRNA2 (Table A2).

### 2.4. Single-Cell Clustering

To categorize the dynamic responses of the cells to the first and the second stress into different classes, we compared the maximum levels of fluorescence reached during the first and second stress, while taking the basal fluorescence level prior to the corresponding stress as a reference. After fitting to a three-degree polynomial curve, the ratio between the maximum fluorescence amplitudes of the first and the second stresses was determined. Cells categorized into profile 1 had a ratio over 1, cells categorized into profile 2 had a ratio less than 1 and cells displaying profile 3 had a ratio equal to 1. Cells with a maximum amplitude equal to 0 (no expression) during the second or first stress were categorized separately. All ratios were established with a 5% threshold.

### 2.5. Microfluidics 

An H-shaped microfluidic device created using soft lithography techniques was used to confine the yeast cells in 3.7 µm-high channels. Hyperosmotic stress was induced using SC medium containing 2% glucose supplemented with 1 M sorbitol. The media was flowed into the microfluidic chip using a peristaltic pump (ISMATEC, Wertheim, Germany) at a flow rate of 120 µL/min. 

### 2.6. Transcriptional Inhibition 

Cells were exposed to SC containing thiolutin (ab143556; Abcam, Cambridge, UK) at final concentration of 50 µg/mL (diluted in DMSO) for 1 h prior to and during stress. SC medium was flowed over the cells using a peristaltic pump at a flow rate of 120 µL/min for 4 h to wash thiolutin out of the microfluidic device. 

### 2.7. Microscopy

Yeast cells were observed using a UplanFLN 100×/1.3 Oil Ph3 Ul2 objective and inverted Olympus IX71 microscope (Olympus, Tokyo, Japan). Images were recorded using a Cool Snap HQ2 camera (Princeton Instruments, Thousand Oaks, CA, USA). All experiments were conducted at 30 °C. Yeast were imaged every 5 min with 20 ms exposure in bright light and 200 ms in fluorescent light. The microscope was controlled using MicroManager open source software interfaced with Matlab (MathWorks, Natick, MA, USA).

## 3. Results

### 3.1. The Reduced Response of a Population of Cells to Successive Hyperosmotic Stress Suggests Cellular Memory

A population of growing yeast cells in a microfluidic device was submitted to short, repeated hyperosmotic stresses using 1 M sorbitol (Figure 1A). To measure the response to hyperosmotic stress, we created a reporter of HOG pathway activity by tagging the pSTL1 promoter with the fluorescent protein yECITRINE (yEFP) to enable quantification of fluorescence at the single-cell level as a function of time [28] (Figure 1B). Individual cells could be tracked during the time course of an experiment, allowing us to quantify the activation of pSTL1 in response to the first and second osmotic stresses due to the short, transient activation of the HOG pathway (Figure 1C). The limited duration of the stress (8 min) and the long delay between the first and second stress (4 h) guaranteed (i) investigation of the genetic response to hyperosmotic stress before adaptation was established 15–30 min after stress [29], and (ii) full recovery of the cells from the first stress as further evidenced by the similar division time after the first and the second stress (not shown), thus allowing us to compare the dynamics of the responses to the first and second osmotic stresses. Moreover, daughter cells born between the two stresses were not considered, as they were not exposed to the first stress and may blur the stress response. We focused exclusively on the population that received both the first and second stresses (Figure 1C). For this population, we calculated the mean fluorescence peaks reached during the first and second stresses, taking the basal fluorescence levels prior to the corresponding stress as a reference. Peak fluorescence reached 77.14 ± 6.71 (a.u.) after the first stress and 55.44 ± 4.26 (a.u.) after the second stress, indicating an average decrease in fluorescence intensity of 20% after the second stress (Figure 1D). The time required to reach peak fluorescence was not different for the first and second stresses (Figure 1E). The decrease in peak fluorescence amplitude between the first and second stresses correlated with a decrease in yEFP protein expression, as detected by Western blotting (Appendix A), suggesting the changes in peak fluorescence were not related to photo-bleaching.

Collectively, these observations suggest the existence of memory of the first stress event at the level of the stressed cellular population. Moreover, the decrease in fluorescence intensity after the second stress appeared to be related to a reduction in protein translation/expression, rather than a shortened duration of transcription events. 

### 3.2. Most, but Not All, Cells Show Cellular Memory at the Single-Cell Level

The memory effect was not shared equally among cells. Indeed, single-cell analysis revealed dynamic variability in the responses of individual cells to the repeated stresses. We classified the single-cell fluorescent trajectories into typical behaviors based on the responses to the first and second stresses (Figure 2A). The most common behavior (55 ± 11%) was a population memory effect, in which the cells exhibited lower fluorescence intensity after the second stress (Figure 2B). However, 18 ± 7% of cells displayed the opposite behavior, with a stronger response after the second stress (Figure 2B). Very few cells showed similar responses to both stresses. Interestingly, we also observed two subpopulations of cells that did not respond to one of the stresses (Figure 2B), although we confirmed that these cells actually perceived the stress by observing transient stress-induced cell shrinkage (Appendix A, Appendix A, Appendix A). 

Taken together, these results show that the population behavior overlooks a richer set of dynamic single-cell responses, which are likely to reflect trace variability in the activation of pSTL1 in response to hyperosmotic stress. 

### 3.3. Cellular Memory Overcomes the Stochasticity of Gene Expression on Average

To determine the importance of intrinsic variability in the different dynamic behaviors shown in Figure 2A, namely the stochastic nature of pSTL1 activation, we performed stochastic simulations based on the Gillespie algorithm [30]. In this algorithm, we modeled gene expression by a stochastic equation and we computed statistically exact solutions to the equation (Table A3 and Table A4). We simulated the transcription of pSTL1 and translation of the fluorescent reporter for 1000 cells submitted to two 8 min stresses separated by 4 h (Figure 2C). The rates of mRNA and protein production and degradation were set as established previously [31] (Table A3). Such a model implies that the computed cells will inevitably respond to both stresses. However, our experiments showed that cells exposed to 8 min stress do not necessarily respond to the stress (Figure 2D), in contrast to cells submitted to a longer stress (Figure 2E). Specifically, the absence of a response disappears as the duration of stress increases; 80% of cells showed activation of pSTL1 in response to 8 min of stress and 100% of cells responded to 1 h of stress (Figure 2F). This experimental observation suggests the existence of a critical time-point, at which all cells will eventually respond to the stress. A stochastic time of activation for the *STL1* promoter was therefore added to the model. As expected in such a memory-free system, cells clustered equally in the two main categories obtained experimentally, and the population did not display any memory effect. Notably, including a transcriptional delay in the model led to the appearance of clusters 4 and 5 (Figure 2G). 

The differences between the clusters observed *in vivo*, and in the simulation suggest that a biological mechanism other than transcriptional and translational noise is responsible for the memory effect. 

### 3.4. Memory of Pulsed Hyperosmotic Stress Does Not Require De Novo Protein Synthesis during the Stress

In order to investigate the biological origin of the memory effect, we first determined if the memory effect was linked to the synthesis of one or more long-lived proteins during the first episode of stress. To test this hypothesis, we inhibited transcription during stress using thiolutin, a well-studied antibiotic that inhibits all three yeast RNA polymerases in a reversible manner [32] (Figure 3A,B). As expected, treatment with thiolutin led to the loss of pSTL1 activation in response to stress: none of the cells treated for 1 h with thiolutin (50 µg/µL), and then subjected to 8 min hyperosmotic stress showed a fluorescent signal whereas 80 ± 20% of cells exhibited a fluorescent signal in response to hyperosmotic stress in the absence of thiolutin (Figure 3A,B). Next, we examined the ability of the cells to respond to hyperosmotic stress after treatment with thiolutin (Figure 3C,D). After 1 h thiolutin treatment, the inhibitor was washed out for 4 h and then the cells were exposed to 8 min hyperosmotic stress. At the population level, a similar response was observed compared to the first stress response in the absence of thiolutin treatment (Figure 3C,D). However, pretreatment with thiolutin slightly reduced the magnitude of the stress response. The maximum intensity after 8 min of stress was 88, 55 ± 7.7% for thiolutin-pretreated cells compared to 100 ± 4.74% for non-treated cells, suggesting that the effect of thiolutin was not completely erased. We then treated cells with thiolutin for 1 h, including the time period of the first stress, washed the inhibitor out for 4 h and then submitted the cells to a second 8 min stress (Figure 3E,F). Under these conditions, thiolutin-pretreated cells showed a marked decrease of 40% in the YFP signal intensity of the second stress response, which is comparable to the 47% decrease in the response to the second stress in cells without thiolutin treatment. 

Consequently, this experiment suggests that the memory effect is not primarily driven by de novo synthesis of proteins during the first stress that help the cells respond to the second stress. To explain the observed memory effect, we hypothesized that the first stress induces chromatin modifications independently of RNA polymerase activity, and that these modifications subsequently affect transcription events at the pSTL1 locus. Thus, it is plausible that chromatin marks would appear in most cells during the first stress and alter the response dynamics of the cells to the second stress. 

### 3.5. Chromosome Position Influences the Dynamic Activity of pSTL1

The *STL1* locus is located on the right arm of chromosome IV in the subtelomeric region, which is prone to silencing under non-stress conditions. To investigate the influence of the chromatin context on the dynamics of pSTL1 activation, we moved the region containing the *STL1* promoter and yECITRINE fluorescent reporter to a distinct centromeric chromatin domain (Figure 4A). The displaced DNA region included the 1 Kb upstream of the *STL1* locus, to ensure a fully functional *STL1* promoter [33]. To compare the activity of pSTL1 in its endogenous position and the centromeric position, we subjected both strains to 2 h hyperosmotic stress and used flow cytometry to quantify fluorescence at several time-points. The activity of centromeric pSTL1 was significantly lower than that of wild-type cells in two independent clones (Figure 4B and Appendix A), even though the integrity of the promoter was preserved [33]. 

Next, the patterns of the consecutive responses to two 8 min hyperosmotic stresses separated by 4 h were compared for endogenous and displaced pSTL1. Cells expressing the *STL1* promoter at the centromeric position exhibited more uniformly distributed responses across the five defined clusters, with a decrease in the proportion of cells displaying the memory effect (from 55 ± 11% to 28 ± 4%, Figure 4C), which is compatible with a purely stochastic process. This result indicated that the chromatin environment may affect the dynamic transcriptional activity of pSTL1. Although a functional pSTL1 promoter was displaced, subtelomeric regulatory elements could have potentially been lost during gene displacement. We ruled out this hypothesis using a Crispr/dCas9VPR system [34] to bypass potential regulatory elements and force the activation of pSTL1 under non-stress conditions (Appendix A). We designed guide RNAs to target Crispr/dCas9VPR within the 1 kb sequence of the displaced pSTL1, and successfully induced fluorescent reporter expression independently of any stress at both the peri-centromeric and subtelomeric positions. However, we still observed a decrease in the activity of pSTL1 at the peri-centromeric position compared to the endogenous position, meaning that the observed differences in pSTL1 expression were not linked to the sequence of the promoter itself or the presence of regulatory elements, but rather to the chromatin environment.

Taken together, our results indicate that the chromatin environment affects the variability of single-cell dynamic responses to short-term stress, and consequently confers cellular memory. 

## 4. Discussion

In the current study, we investigated how individual yeast cells dynamically behave in response to short pulses of hyperosmotic stress. Focusing our study on short stresses allowed us to exclusively analyze the genetic response to repeated hyperosmotic stress and to probe the cell-cell variability that originates from transcriptional events. 

In-depth single-cell analysis revealed yeast exhibit various behaviors in response to two repeated stresses, and we clustered these behaviors into several typical profiles. The response to the second stress could be similar, higher or lower than the response to the first stress. A reduced response to the second stress was the most common behavior and this was named the memory effect. We also considered two additional profiles for cells that did not respond to one of the two stresses. Simulations indicated that the two non-response profiles may depend on transcriptional delays and were validated by the experimental observation that all cells responded to a longer duration of stress, as previously observed [35]. Using stochastic simulations, we established that the five response profiles could be explained by gene expression stochasticity. However, the single-cell quantifications obtained within such a model did not account for the prevalence of the memory effect, indicating that the memory effect is not a reflection of gene expression stochasticity alone.

Studies on cellular memories in response to repeated stresses in budding yeast typically describe more rapid dynamics of gene expression or lower amplitude responses as two possible responses [6,8,36]. Similar to the responses to longer hyperosmotic stress, we observed a decrease in the amplitude of the pSTL1 response between the first and second short pulsed stresses, without any difference in the timing of reactivation [6]. We speculate that the diminished responses to repeated stress could be a strategy to reduce the cellular burden, compared to the demands of invoking similar transcriptional responses to repeated stresses. 

This raises the question of the mechanism that drives the diminished response to repeated hyperosmotic stress. It is possible that—as observed for the response to hyperosmotic stress triggered by NaCl, [6]—a long-lived protein induced by the first stress could remain in the nucleus and inhibit or limit activation of the promoter of interest upon the second stress. However the assumption that long-lived proteins determine the response to repeated osmotic stress seems unlikely because within 3 min after stress, Hog1 is phosphorylated and translocates into the nucleus to up-regulate stress response genes, including *STL1* and is rapidly exported from the nucleus within 15–30 min after the start of stress [25]. Furthermore, transcriptional inhibition experiments showed the memory effect does not seem to require de novo protein synthesis. Since the transcriptional inhibitor thiolutin inhibits all three RNA yeast polymerases, and thus de novo transcription, but does not prevent potential transcription factors from binding to promoter sequences, specific histone marks may possibly be left after exposure to the first stress. Those marks could serve as traces of previous promoter activity and could explain the diminished responses to a subsequent stress. This interpretation awaits validation using single-cell ChIP assays. 

We further showed that the distribution of dynamic variability between single cells was dependent on the chromosomal position of the pSTL1 locus. When displaced to a pericentromeric domain, pSTL1 showed decreased activity in response to the second stress. We confirmed this effect was not due to potential loss of regulatory sequences since we showed that activation of pSTL1 by a CrispR-dCAS9-VPR construct—which bypasses the need for stress factors to invoke a response—still depended on the genomic position. At the displaced pericentromeric position, the stochasticity of gene expression prevails and the memory effect is lost. Consequently, an active mechanism may occur in subtelomeric regions in response to stress, which allows the memory effect to become predominant within the cell population. Interestingly, it has already been observed that changing the position of a gene in the genome can alter its expression [37]. It was proposed that these alterations were due to changes in transcriptional noise or the noisy steps when cells transition between two expression levels [37]; the latter has been shown to enhance cellular memory in cells subjected to glucose limitation stress [21].

One key difference between the two genomic positions we analyzed is the variation in the amplitude of the pSTL1 response. Thus, we propose that transcriptional marks or transactivators induce high-level transcriptional activity during the first stress, which overcome the stochasticity of gene expression and lead to the emergence of cellular memory. A parallel can be drawn with previous studies performed in *B. subtilis*, in which transcriptional events occurring above a certain threshold have been described to lead to the emergence of cellular memory [38]. Although bacteria are prokaryotes, similarities may exist between *B. subtilis* and yeast in regards to the biology of memory.

It could be hypothesized that high activity of a promoter of interest could mean an open chromatin structure. Although marks of acetylation are usually associated with a high level of gene expression, the activity of pSTL1 reduces in the absence of the histone deacetylase Rpd3 [39]. In the experimental context studied here, marks of deacetylation may possibly be involved in high-level transcriptional activity. Thus, it would be interesting to investigate the potential role of (de)acetylation by, for instance, forcing a high level of (de)acetylation during the first stress only. 

Stochastic gene expression leads to behavioral diversity. From an evolutionary point of view, this diversity of the responses to repeated stresses allows selection of the most adapted response. Under our experimental conditions, the preference towards a memory effect suggests the specific subtelomeric position of pSTL1 offers the optimal level of regulation for improved adaptation. 

Overall, our work shows how single-cell studies are critical to the analyses of stress memory. Furthermore, our experiments indicate that the establishment and transmission of memory do not require exposure to long-term stress and can be induced by short, pulsed stresses. This work could serve as a basis for broader studies of the genomic positioning of stress response genes in budding yeast in response to fluctuating environments.

## Figures and Tables

**Figure 1 cells-08-00582-f001:**
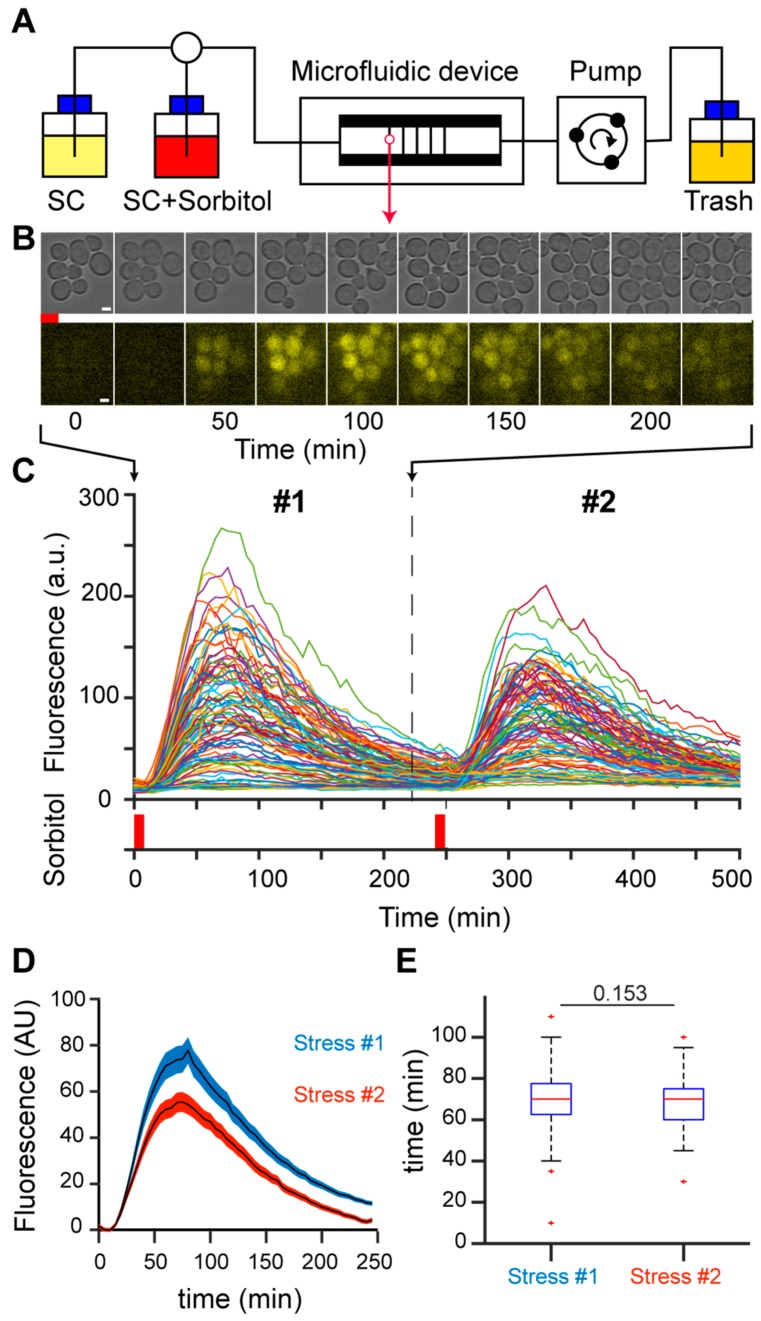
(**A**) Experimental setup. We used multi-layer H-shaped microfluidic devices composed of two large 50 µm-high and 40-µm thick flow channels, with 400 µm × 400 µm × 3.7 µm observation chambers. Cells are trapped in the chambers and grow as a monolayer, which facilitates cell segmentation and tracking. The medium flowed through the channels diffuses into the chambers. Hyperosmotic stress-activated expression of pSTL1-yECRITRINE was triggered by exposure to 1 M sorbitol for 8 min. SC, synthetic complete medium. (**B**) Representative images of cells in the microfluidic chamber exposed to 8 min stress and left to recover for 4 h (240 min), scale bar, 5 µm. Cells were imaged every 5 min in bright light (20 ms exposure, upper row) and fluorescent light (200 ms exposure, lower row). (**C**) Fluorescent signals for individual cells exposed to two successive 8 min-stresses separated by 4 h. Hyperosmotic stress is indicated by the red bars, # 1 and # 2 indicate first and second stress, respectively. (**D**) Fluorescence responses of a population of *n* = 97 cells. The mean (± standard error of the mean) responses of cells to a first stress (blue), followed by a second stress 4 h later (red) are represented. Fluorescence levels were normalized to the fluorescence value before each corresponding stress. The fluorescence peak decreased from the first to the second stress. (**E**) Analyses of the temporal responses to two consecutive stresses in the same population. The duration between stress induction and peak fluorescence was similar for the first and second stresses.

**Figure 2 cells-08-00582-f002:**
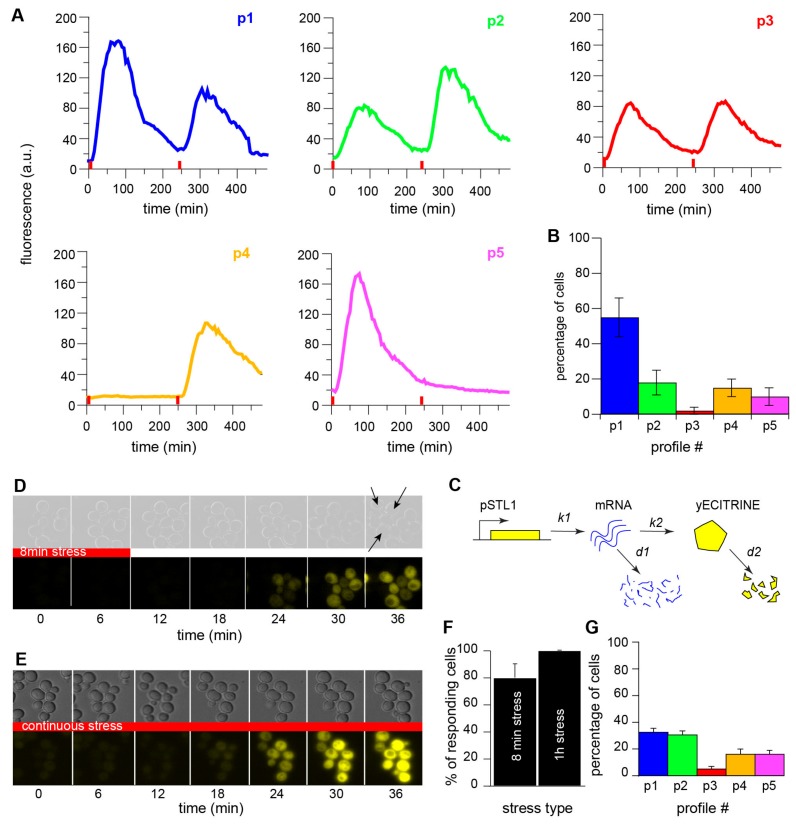
(**A**) Examples of the five typical single-cell response profiles. Although the single-cell analysis revealed dynamic variability in the responses of single cells, we defined five typical response profiles (p1 to p5). (**B**) Single-cell clustering. Based on peak fluorescence values, the dynamic variability of the responses was clustered into five typical fluorescence responses. Errors bars represent the standard errors. A total of 708 cells were analyzed in three independent experiments. (**C**) Modeling of gene expression upon stress in a memory-free system. Stochastic simulations with the Gillespie algorithm were used to model transcription of the fluorescent reporter and protein translation upon stress. (**D**) Time series of images of cells subjected to 8 min stress. The arrows on the last bright field image show the cells that did not respond to the stress. (**E**) Time series of images of cells subjected to continuous stress. All cells responded to the stress. (**F**) Quantification of responsive cells. After exposure to 8 min stress, 80% of cells showed a response, whereas 100% of cells responded to 1 h stress. (**G**) Single-cell quantification of computed cells according to the five typical response profiles. In this case, the model includes a randomly selected transcription delay of between 0 and 10 min for each computed cell. This delay was also varied for the two stresses. The simulation was run twice and the peak fluorescence values were used to cluster the responses of the computed cells according to the five typical response profiles.

**Figure 3 cells-08-00582-f003:**
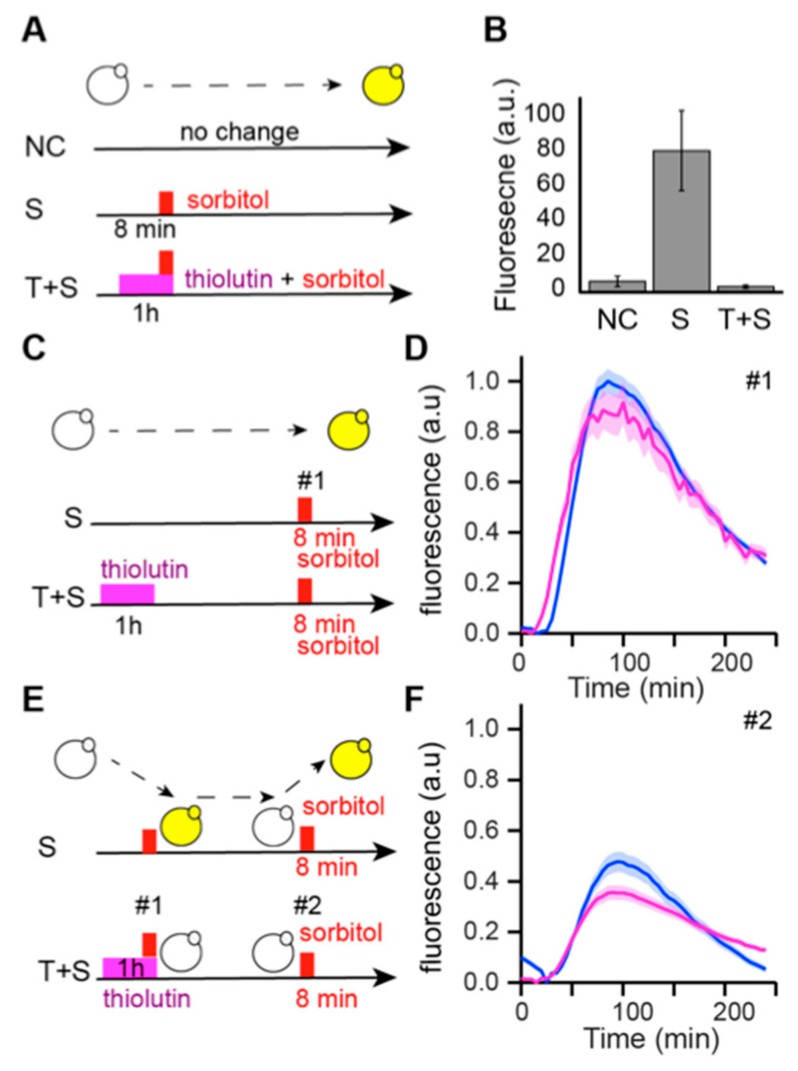
(**A**) Principle of the transcription inhibition experiment using thiolutin. Non-stressed cells received no sorbitol (NC) and stressed (S) cells were subjected to 8 min hyperosmotic stress (small red rectangle). (T + S) cells were treated with thiolutin for 1 h (pink rectangle) prior to and during stress. (**B**) Single-cell quantification of cellular fluorescence after 70 min under non-stress conditions (NC), hyperosmotic conditions (S) and hyperosmotic conditions in the presence of thiolutin (T + S). (**C**) Principle of the experiment to control thiolutin wash out. (Upper row). (S) cells were cultured in the microfluidic device for 4 h before being exposed to 8 min hyperosmotic stress. (Lower row) (T + S) cells were treated with the transcriptional inhibitor thiolutin for 1 h, thiolutin was washed out for 4 h and then cells were exposed to 8 min hyperosmotic stress. (**D**) pSTL1 fluorescence responses of cells treated (*n* = 101, pink) or untreated (*n* = 97, blue) with thiolutin after stress; similar responses were observed. Fluorescence levels were normalized to the peak fluorescence value of non-treated cells. (**E**) Principle of quantification of the memory effect in the presence of thiolutin. (S) cells were exposed to two 8 min hyperosmotic stresses 4 h apart. (T + S) cells were treated with thiolutin for 1 h and exposed to 8 min hyperosmotic stress, thiolutin was washed out for 4 h and cells were then exposed to a second 8 min stress. (**F**) Population quantification of stress memory for cells subjected to a first stress in the absence (blue) or presence of thiolutin (*n* = 101, pink). Fluorescence levels were normalized to the peak fluorescence value of the non-treated cells presented in (**D**).

**Figure 4 cells-08-00582-f004:**
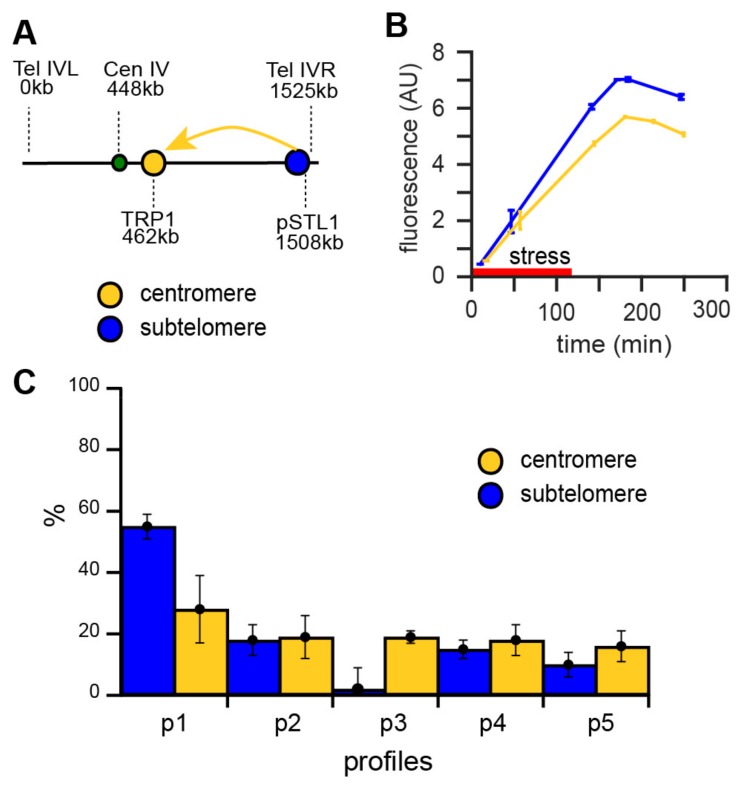
(**A**) Displacement of pSTL1 towards the peri-centromere of chromosome IV. Sketch of the endogenous genomic position of pSTL1 on chromosome IV. The promoter was moved to the *TRP1* locus on the same chromosome. Genomic positions are indicated in Kb. (**B**) Reduced activity of displaced pSTL1 in response to stress. Fluorescence quantification of promoter activity in response to 2 h hyperosmotic stress (red bar) for the endogenous promoter (blue) and displaced promoter (yellow). Data are mean ± standard deviation of triplicate experiments. (**C**) Displacement of pSTL1 leads to loss of the memory effect. Single-cell quantification of cells containing the displaced pSTL1 promoter in response to two hyperosmotic stresses. Response profiles were classified according to the five typical profiles in Figure 2.

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
