# Peer review of "Hyperosmotic Stress Response Memory is Modulated by Gene Positioning in Yeast"

_cells, 2019, doi:10.3390/cells8060582_

Round 1

Reviewer 1 Report

The dominant mechanism by which cells respond to environmental stresses is through transcriptional regulation. In some cases, the nature of the response is altered by previous stimuli, a phenomenon called memory. The authors explore a type of memory in yeast cells whereby previous osmotic stress leads to a diminished response to the same stimulus several hours later. Using microfluidics to allow single, live yeast cells to be followed, the authors find that five different patterns are observed at distinct frequencies. The largest fraction of cells (~50%) show a diminished response to the second stimulus (p1). However, smaller fractions respond equally/more strongly and some cells fail to respond to one or the other stimulus (p2-5). This behavior could not be modeled using a simulation for transcriptional noise, suggesting that it represented another mechanism.

When the authors used thiolutin to inhibit transcription during the first stress, the diminished response was still observed, arguing against the idea that perdurance of protein from the first response was not responsible. In contrast, when the reporter was moved from a sub-telomeric location to a peri-centromeric location, expression was decreased and the memory response was qualitatively altered, leading to approximately equal representation in each of the five classes. The authors conclude that the genomic context impacts this memory phenomenon, perhaps through chromatin structure.

The observations reported in this manuscript are intriguing and carefully analyzed. However, to strengthen the conclusions, I would suggest the following limited number of critical experiments:

1. Test the effect of sir2Δ on the memory phenomenon. This will establish if STL1 memory is critically dependent on subtelomeric silencing.
2. Test the effect of htz1Δ on the memory phenomenon. This will test if STL1 memory is dependent on (or modified by) boundary activity at subtelomeres, which is dependent on H2A.Z. Together with point #1, these results will more rigorously test the model that chromatin structure is important for memory.
3. Introduce pSTL1-yECITRINE-HIS5 at another subtelomeric locus to test if subtelomeric localization is sufficient to recapitulate proper regulation.

Minor comments/corrections:

1. In several places (lines 136, 212, 236), the authors use the phrase “cells were submitted to stress”. A better phrasing would be, “cells were subjected to stress”.
2. Line 153, the word “picks” should be replaced with “peaks”.
3. Lines 157-159: the authors state, “Moreover, the decrease of fluorescence intensity at the second stress is seemingly due to a reduction of protein production rate rather than a shortened duration of transcription events.” It was unclear to me why they make this statement. Please clarify.
4. Line 211: Please explain the Gillespie algorithm in more detail.
5. Line 352: the phrase “noise, or a in the noisy steps” should read “noise, or in the noisy steps”.
6. Figures 1B, 2C & 2E should include scale bars.
7. Figure 3: please take care to label the conditions more clearly. S and T+S mean different things in panels A, B and C. This should be avoided.
8. Figure 3D & F: identify the blue and pink curves.
9. Figure 3D & F: the meaning of #1 and #2 is obscure. Please define or remove.
10. Figure 4: The red bar indicating the stress is the same color as the curve for the cenproximal pSTL1; use a different color so that it can be properly distinguished in the legend.

Author Response

The dominant mechanism by which cells respond to environmental stresses is through transcriptional regulation. In some cases, the nature of the response is altered by previous stimuli, a phenomenon called memory. The authors explore a type of memory in yeast cells whereby previous osmotic stress leads to a diminished response to the same stimulus several hours later. Using microfluidics to allow single, live yeast cells to be followed, the authors find that five different patterns are observed at distinct frequencies. The largest fraction of cells (~50%) show a diminished response to the second stimulus (p1). However, smaller fractions respond equally/more strongly and some cells fail to respond to one or the other stimulus (p2-5). This behavior could not be modeled using a simulation for transcriptional noise, suggesting that it represented another mechanism.

When the authors used thiolutin to inhibit transcription during the first stress, the diminished response was still observed, arguing against the idea that perdurance of protein from the first response was not responsible. In contrast, when the reporter was moved from a sub-telomeric location to a peri-centromeric location, expression was decreased and the memory response was qualitatively altered, leading to approximately equal representation in each of the five classes. The authors conclude that the genomic context impacts this memory phenomenon, perhaps through chromatin structure.

The observations reported in this manuscript are intriguing and carefully analyzed. However, to strengthen the conclusions, I would suggest the following limited number of critical experiments:

1.     Test the effect of sir2Δ on the memory phenomenon. This will establish if STL1 memory is critically dependent on subtelomeric silencing.

Deletion of Sir2 will change the redox balance within the cell. This can compromise biochemical reactions and cause additional effects that may compromise our understanding of the phenomenon. However, as suggested by the referee, we questioned the effect of subtelomeric silencing by testing the impact of the SIR complex through Sir3 deletion. Strain deletion was molecularly verified by PCR and physiologically verified, by the inability of sir3∆ strain to mate, because inefficient silencing of mating type locus by the SIR complex. To our surprise, a very moderate effect in the transcriptional activity of pSTL1-YFP was observed in sir3∆ strain using FACS (left figure).  Accordingly, single-cell experiments show no impact of sir3∆ (right figure). Thus STL1 memory does not seem to be critically dependent on subtelomeric silencing by Sir3. This observation is however consistent with a study from Mazor and Kupiec (Developmentally regulated MAPK pathways modulate heterochromatin in Saccharomyces cerevisiae. Nucleic Acids Res. 2009. 37, 4839-49) showing that the Sir complex is depleted from the subtelomeric chromatin upon hyperosmotic stress. As a result, stress response genes (located at subtelomeres and submitted to the Sir silencing) are no longer subject to silencing and can be expressed. This could explain why there is no effect of the Sir complex on the memory effect.

Figure should be seen in the word file

Legend to the figure (Left) No effect of the Sir complex on the transcriptional activity of pSTL1. Fluorescence quantification of promoter activity in response to 2 h hyperosmotic stress for the endogenous promoter (blue) and the sir3∆ strain (green). (Right) No impact of the Sir complex on the memory effect. Single-cell quantification of sir3∆ cells in response to two hyperosmotic stresses. Response profiles were classified according to the five typical profiles in Figure 2. Control with cells expressing pSTL1 at the endogenous position (blue).

2. Test the effect of htz1
Δ on the memory phenomenon. This will test if STL1 memory is dependent on (or modified by) boundary activity at subtelomeres, which is dependent on H2A.Z. Together with point #1, these results will more rigorously test the model that chromatin structure is important for memory.

Single-cell experiments were performed on Htz1 (which encodes H2A.Z) deletion and showed that they have no impact on the memory effect, although a great variability between cells was observed. However, the reason we have studied these genes in particular was to test if the memory effect (decrease of amplitude of response) could actually be a reflection of the chromatin being less opened with the stress. Therefore, forcing a hyperacetylated chromatin would allow defining a putative chromatin cause for memory. This was not the case in our experiments. We are however convinced that such analyses deserve further work beyond the scope of this study.

3. Introduce pSTL1-yECITRINE-HIS5 at another subtelomeric locus to test if subtelomeric localization is sufficient to recapitulate proper regulation.

We have tried to investigate if subtelomeres are equivalent so that the regulation would be recovered at another subtelomere. However, yeast subtelomeres share a high degree of sequence similarity (for instance see Pryde FE & Louis EJ (1999) Limitations of silencing at native yeast telomeres. EMBO J 18: 2538–2550) that might have hampered our attempts to displace the promoter of interest at other subtelomeres.

Minor comments/corrections:

1. In several places (lines 136, 212, 236), the authors use the phrase “cells were submitted to stress”. A better phrasing would be, “cells were subjected to stress”.

After English editing this is now corrected.

2. Line 153, the word “picks” should be replaced with “peaks”.

After English editing this is now corrected.

3. Lines 157-159: the authors state, “Moreover, the decrease of fluorescence intensity at the second stress is seemingly due to a reduction of protein production rate rather than a shortened duration of transcription events.” It was unclear to me why they make this statement. Please clarify.

We have clarify the statement as following: ”The decrease in peak fluorescence amplitude between the first and second stresses correlated with a decrease in yEFP protein expression, as detected by Western blotting (Figure A1), suggesting the changes in peak fluorescence were not related to photo-bleaching”.

4. Line 211: Please explain the Gillespie algorithm in more detail.

We have clarified the statement in the text and added system equations in Table A2 .

5. Line 352: the phrase “noise, or a in the noisy steps” should read “noise, or in the noisy steps”.

After English editing this is now corrected.

6. Figures 1B, 2C & 2E should include scale bars.

We have included scale bars.

7. Figure 3: please take care to label the conditions more clearly. S and T+S mean different things in panels A, B and C. This should be avoided.

We have better explained in the legend to what S+T refers.

8. Figure 3D & F: identify the blue and pink curves.

This is now identified in the legend.

9. Figure 3D & F: the meaning of #1 and #2 is obscure. Please define or remove

This is now defined in the legend.

10. Figure 4: The red bar indicating the stress is the same color as the curve for the cenproximal pSTL1; use a different color so that it can be properly distinguished in the legend
.

            We changed the color of the cen proximal pSTL1 to yellow.

Reviewer 2 Report

The manuscript is very well written and the idea to study aspects related to memory of unicellular organisms regarding the exposure to osmotic stress is very interesting. 

The authors clearly demonstrate that cells exposed to a second stress exhibit a lower response, attributed to some form of cellular memory. Nevertheless, the authors do not take into consideration the possibility that the time between the two stress pulses being rather high (i.e., 4h), some loss in cell fitness/viability might be actually responsible for the lower response to the second stress pulse. Some data/comments in this regard may increase the credibility of an otherwise very solid study.   

Author Response

The manuscript is very well written and the idea to study aspects related to memory of unicellular organisms regarding the exposure to osmotic stress is very interesting

We thank the referee for these very positive comments highlighting the interest of our study.

The authors clearly demonstrate that cells exposed to a second stress exhibit a lower response, attributed to some form of cellular memory. Nevertheless, the authors do not take into consideration the possibility that the time between the two stress pulses being rather high (i.e., 4h), some loss in cell fitness/viability might be actually responsible for the lower response to the second stress pulse. Some data/comments in this regard may increase the credibility of an otherwise very solid study. 

The referee raises a question that has concerned us as well. When displaced at the centromere, there is no memory effect, suggesting that the memory effect is independent from a loss in cell fitness and rather on the transcription level. Yet, we have compared the doubling time after the first and the second stress and found it to be similar  (t1/2 after first stress = 99 min, t1/2 after second stress= 100 min, left graph below). A similar doubling time s observed when pSTL1 is displaced in the pericentromeric domain (right graph). We have now added this values and comment in the main text, first paragraph.

Figure should be seen in the uploaded word file

Legend to the figure. Analysis of the division time of the same population upon two successive stress. (Left) Division time of N=100 cells with pSTL1 at the subtelomeric position upon first (mean 98.02 min) and second stress (mean 99.55) (p=0.2774). (Right) division time of N=100 cells with pSTL1 at the pericentromeric position upon first (mean 100.65min) and second stress (100.85min) (p=0.8917).

Reviewer 3 Report

Meriem et al, have presented interesting data that stress effect single cell memory and its association with chromatin structure. The study has potentially interesting new data, but the manuscript requires improvement in organization and clarity.

Authors have mentioned about NaCl stress in discussion, does it will have same effect on cellular memory? Does NaCl have mechanism of action as glycogen?

Different factors effecting cellular memory, like age and shape. Why they did not perform the experiment on synchronized cell, it could be more informative.

Author Response

Meriem et al, have presented interesting data that stress effect single cell memory and its association with chromatin structure. The study has potentially interesting new data, but the manuscript requires improvement in organization and clarity.

Careful proof-reading and copy editing were now performed by a professional.

Authors have mentioned about NaCl stress in discussion, does it will have same effect on cellular memory? Does NaCl have mechanism of action as glycogen?

Rienzo et al., Mol Cell Biol. 2015. 35, 3669-83, have shown a similar decreased response when cells are exposed to NaCl stress.  In the cited study, cells were pretreated with 0.7M NaCl overnight, before exposure to different salt concentrations. Authors involved a defense protein, the cation efflux protein Ena1, in the acquisition of stress resistance, suggesting that mechanisms involved are different between osmotic stresses.

Different factors effecting cellular memory, like age and shape. Why they did not perform the experiment on synchronized cell, it could be more informative.

We also tried to investigate the influence of the cell cycle by analyzing cell stage concomitantly to stress response by co-expressing histone H2B fused to fluorescent mCherry (Htb2-mCherry) with pSTL1-yEFP. The rationale was to define the cell cycle phase with nuclear Htb2-mCherry signal and correlate strength response to successive hyperosmotic stresses with the phase of the cell cycle. However, our results were not conclusive. We also tried to setup experiments using calcofluor to determine the age of the cells, but we were not successful in calcofluor probing.